# Jarid2 binds mono-ubiquitylated H2A lysine 119 to mediate crosstalk between Polycomb complexes PRC1 and PRC2

Sarah Cooper[1,*], Anne Grijzenhout[1,*], Elizabeth Underwood[1,*], Katia Ancelin[2], Tianyi Zhang[1], Tatyana B. Nesterova[1], Burcu Anil-Kirmizitas[1], Andrew Bassett[3,†], Susanne M. Kooistra[4,5], Karl Agger[4,5], Kristian Helin[4,5,6], Edith Heard[2] & Neil Brockdorff[1]

The Polycomb repressive complexes PRC1 and PRC2 play a central role in developmental gene regulation in multicellular organisms. PRC1 and PRC2 modify chromatin by catalysing histone H2A lysine 119 ubiquitylation (H2AK119u1), and H3 lysine 27 methylation (H3K27me3), respectively. Reciprocal crosstalk between these modifications is critical for the formation of stable Polycomb domains at target gene loci. While the molecular mechanism for recognition of H3K27me3 by PRC1 is well defined, the interaction of PRC2 with H2AK119u1 is poorly understood. Here we demonstrate a critical role for the PRC2 cofactor Jarid2 in mediating the interaction of PRC2 with H2AK119u1. We identify a ubiquitin inter-action motif at the amino-terminus of Jarid2, and demonstrate that this domain facilitates PRC2 localization to H2AK119u1 both *in vivo* and *in vitro*. Our findings ascribe a critical function to Jarid2 and define a key mechanism that links PRC1 and PRC2 in the establishment of Polycomb domains.

[1] Developmental Epigenetics, Department of Biochemistry, University of Oxford, South Parks Road, Oxford OX1 3QU, UK. [2] Institut Curie, CNRS UMR3215, INSERM U934, 26 rue d'Ulm, Paris 75248, France. [3] Genome Engineering Oxford, Sir William Dunn School of Pathology, University of Oxford, South Parks Road, Oxford OX1 3RE, UK. [4] Biotech Research and Innovation Centre (BRIC), University of Copenhagen, 2200 Copenhagen, Denmark. [5] Centre for Epigenetics, Ole Maaløes Vej 5, University of Copenhagen, 2200 Copenhagen, Denmark. [6] The Danish Stem Cell Center (Danstem), University of Copenhagen, Blegdamsvej 3, 2200 Copenhagen, Denmark. * These authors contributed equally to this work. † Present address: Wellcome Trust Sanger Institute, Wellcome Genome Campus, Hinxton, Cambridgeshire CB10 1SA, UK. Correspondence and requests for materials should be addressed to N.B. (email: neil.brockdorff@bioch.ox.ac.uk).

The Polycomb repressive complexes PRC1 and PRC2 play a central role in developmental regulation of the genome in multicellular organisms. Both PRC1 and PRC2 catalyse specific histone modifications, H2A lysine 119 ubiquitylation (H2AK119u1) and H3 lysine 27 methylation (H3K27me3), respectively, and these activities together are critical for Polycomb function at target gene loci[1].

In mammals, Polycomb complexes occupy broad domains that correspond to CpG island promoters of repressed target genes[2], and additionally to the inactive chromosome, present in cells of XX females[3–5], and other atypical loci, including, during early embryogenesis, to pericentric heterochromatin (PCH)[6]. Polycomb targets in most instances are co-occupied by both PRC1 and PRC2. Conventionally, this has been attributed to crosstalk involving the CBX subunit of canonical PRC1 complexes binding to PRC2-mediated H3K27me3 (ref. 7). Accordingly, initiation of Polycomb domain formation has generally been attributed to sequence-specific factors and/or non-coding RNAs that target PRC2 complexes. However, more recent findings demonstrate that PRC1 recruitment also has a role in initiating Polycomb domain formation[8–11], and moreover, that PRC1-mediated H2AK119u1 can direct recruitment of PRC2 complexes[12–14]. A specific PRC2 sub-complex, which includes the cofactors Aebp2 and Jarid2, together with core subunits, has been implicated in recognition of H2AK119u1 (ref. 14), although the molecular mechanism for this interaction is currently unknown.

In this study, we demonstrate using different cell-based models that the PRC2 cofactor Jarid2 mediates interaction of PRC2 with H2AK119u1. We identify a ubiquitin interaction motif (UIM) at the amino-terminus of Jarid2, and demonstrate that this domain is required for PRC2 localization to H2AK119u1-modified chromatin both *in vivo* and *in vitro*. Our findings ascribe a critical function to the Jarid2 protein and additionally elucidate a key molecular mechanism for the recognition of H2AK119u1 by PRC2, furthering our understanding of the link between PRC1 and PRC2 in the establishment of Polycomb domains.

## Results

**Jarid2 mediates recognition of H2AK119u1.** To define the role of the PRC2 sub-complex associated with the cofactors Aebp2 and Jarid2 in recognition of H2AK119u1 *in vivo*, we made use of MBD-RPCD, a fusion protein construct which directs H2AK119u1 to methylated CpG sites, including at PCH domains[13] (Fig. 1a). Expression of MBD-RPCD targeted H2AK119u1 to PCH in wild type, Aebp2 null and Jarid2 null mouse embryonic stem cells (mESCs) (Supplementary Fig. 1a,b), consistent with our previous findings[13], and in wild type and Aebp2 null mESCs, the H2AK119u1 recruited PRC2 (H3K27me3) (Fig. 1b,c). However, in Jarid2 null mESCs, H3K27me3 deposition was abolished (Fig. 1d,e). This result is not due to PRC2 intrinsically losing capacity to catalyse H3K27me3, as direct tethering of the catalytic subunit of PRC2, Ezh2, using an MBD-Ezh2 fusion, did result in H3K27me3 deposition (Supplementary Fig. 1c,d). These results suggest that Jarid2, but not Aebp2, plays a role in recognition of H2AK119u1 by PRC2.

To further test the role of Jarid2 in PRC2 recruitment by H2AK119u1, we used a different model system, Dnmt1 conditional knockout mESCs, in which depletion of DNA methylation triggers deposition of both H2AK119u1 and H3K27me3 at PCH domains[13] (Fig. 2a). Thus, we generated Jarid2 null alleles in Dnmt1 conditional knockout mESCs using CRISPR/Cas9 (Supplementary Fig. 2a), and then induced deletion of Dnmt1 by tamoxifen-mediated activation of CreER. Depletion of DNA methylation from PCH (and genome wide) occurred

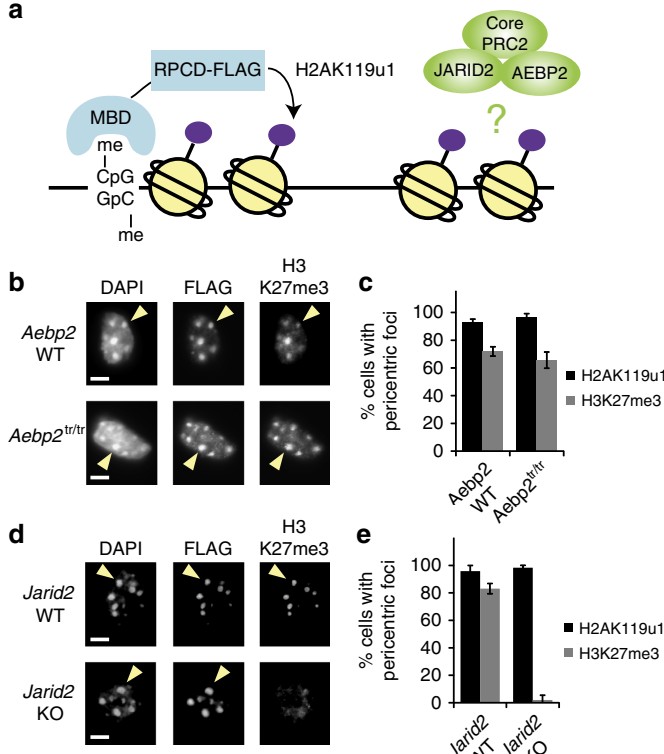

**Figure 1 | JARID2 is required for recruitment of PRC2 to H2AK119u1 modified chromatin.** (**a**) Schematic of the experiment. The MBD localizes to high meCpG regions including PCH. The minimal E3 ligase fusion protein (RPCD) catalyses H2AK119u1 but does not interact with PRC1 or PRC2. (**b**) Immunofluorescence staining of WT and *Aebp2^{tr/tr}* mESCs. Scale bar, 5 μm. Arrowhead indicates a PCH focus. (**c**) Quantification of H3K27me3-positive PCH domains shown in **b**. (**d,e**) As (**b,c**) but for *Jarid2* WT and *Jarid2* KO mESCs. A minimum of 300 cells were counted in three biological repeats. Error bars indicate s.d.

within 6 days (Supplementary Fig. 2b), and loss of Jarid2 had little or no effect on global levels of H3K27me3 (Supplementary Fig. 2c), consistent with previous reports[15,16]. However, while H3K27me3 deposition at PCH occurred after 6 or 12 days tamoxifen treatment in wild type (WT) mESCs, it was undetectable in the Jarid2 null cells (Fig. 2b and Supplementary Fig. 2d). H2AK119u1 deposition at PCH on the other hand was detected in WT and Jarid2 null mESCs (Fig. 2c). Again, tethering MBD-Ezh2 at PCH resulted in H3K27me3 deposition in WT and Jarid2 null ESCs, demonstrating that Jarid2 null cells retain the capacity to deposit H3K27me3 at these sites (Supplementary Fig. 2e,f). Analysis of Dnmt1 deficient mESC sublines stably transfected with Ezh2-eGFP demonstrated PRC2 localization to PCH in WT but not in Jarid2 null mESCs (Supplementary Fig. 2g). This finding confirms that Jarid2 facilitates PRC2 binding to H2AK119u1 chromatin.

**Jarid2 and PRC2 recruitment to the inactive X chromosome.** In unpublished work, we have found that H2AK119u1 mediated by a variant PRC1 complex, PCGF3/5-PRC1, initiates Xist RNA-dependent Polycomb domain formation in X chromosome inactivation. Interestingly, it has been reported that knock-down of Jarid2 reduces Xist-mediated recruitment of PRC2 in X chromosome inactivation[17]. Together with the findings reported herein, these observations suggest that PRC2 recruitment in X inactivation may also require recognition of

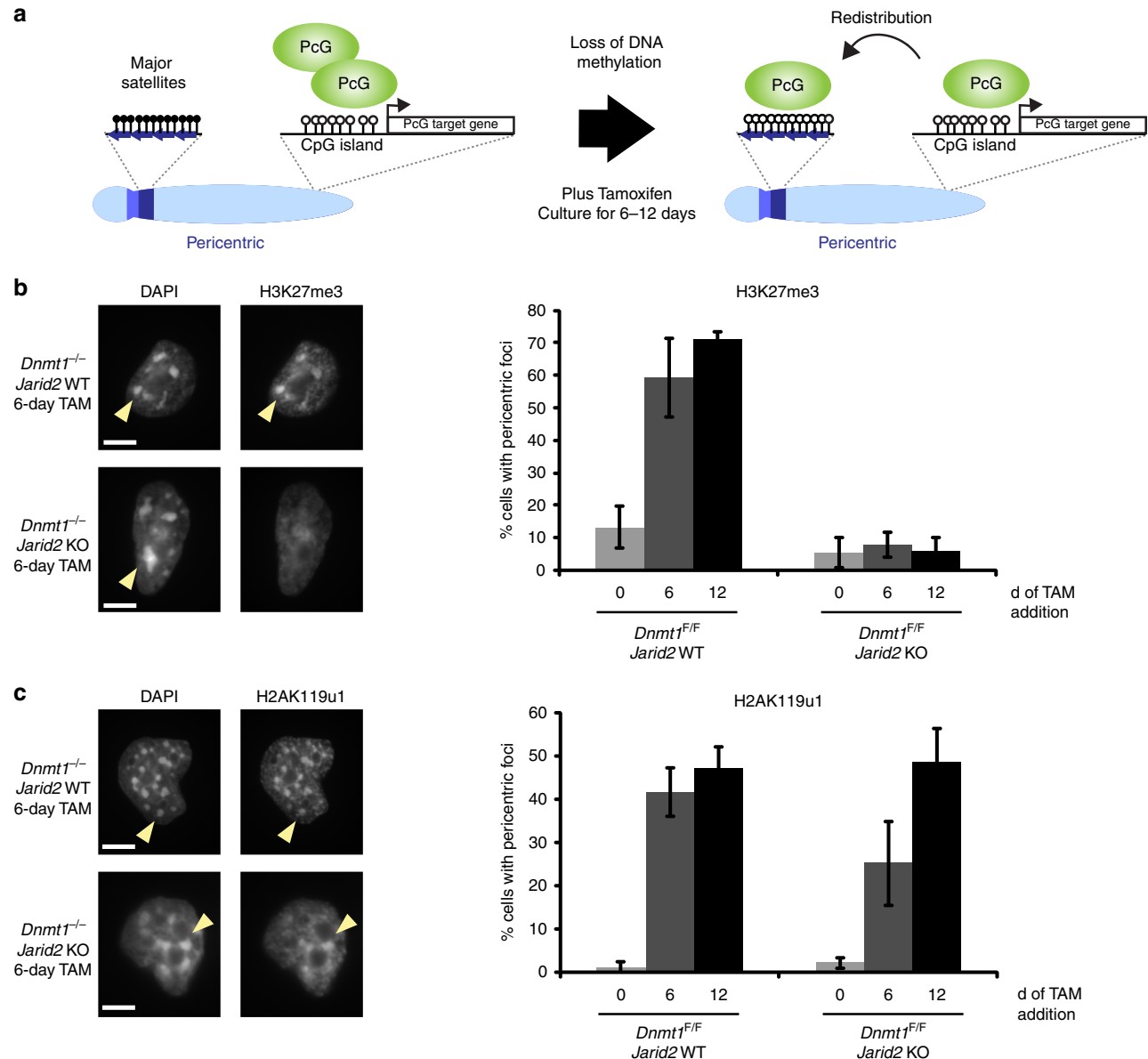

**Figure 2 | JARID2 is required for establishment of H3K27me3 at PCH domains upon loss of DNA methylation.** (**a**) Schematic of the experimental set-up. DNA methylation (filled lollipops) was removed by tamoxifen induction of conditional *Dnmt1F/F* mESCs and culturing for 6–12 days. In response to the loss of DNA methylation PRC1 (H2AK119u1) and PRC2 (H3K27me3) complexes localize to PCH (major satellites). (**b**) Immunofluorescence staining with H2AK119u1 (left) and quantification of PCH domains (right) of *Dnmt1−/−* and *Dnmt−/−* *Jarid2* KO mESCs. (**c**) As (**b**), except staining and quantification for H3K27me3. A minimum of 300 cells were counted in three biological repeats. Error bars indicate s.d. Scale bars, 5 μm.

H2AK119u1 by Jarid2. To test this hypothesis, we used CRISPR/Cas9 to generate null alleles in the mESC cell line, BglXist1, in which an inducible Xist RNA transgene has been engineered with BglG stem loops, enabling tagging with Bgl-mCherry fusion protein[18] (Supplementary Fig. 3a,b). In parallel, we generated null alleles for the core PRC2 protein Suz12, and for the chromatin remodeller ATRX, recently proposed to play a pivotal role in PRC2 recruitment by Xist RNA[19] (Supplementary Fig. 3c). We then assayed for the presence of PRC2 (Ezh2 and Suz12) and H3K27me3 domains after inducing Xist RNA expression in wild-type and mutant mESC lines. Examples and quantitative analysis are shown in Supplementary Fig. 3d,e. In Suz12 null BglXist1 mESCs, PRC2 localization and H3K27me3 was largely abolished, consistent with expectations. Deletion of Jarid2 strongly reduced PRC2

recruitment, consistent with published findings[17], although in contrast to PCH domains, residual activity was observed, evidenced by the presence of H3K27me3 domains in a proportion of cells. Surprisingly, deletion of the *Atrx* gene had no effect on either PRC2 or H3K27 domains formed in response to Xist RNA expression. This latter observation argues against the proposal that ATRX is important for PRC2 recruitment by Xist RNA[19].

We further defined the contribution of Jarid2 to PRC2 recruitment by Xist RNA in preimplantation embryos from a Jarid2 conditional knockout mouse line, employing the ZP3-CRE oocyte-specific driver[20] to deplete both maternal and embryonic Jarid2 (Supplementary Fig. 4a,b,c). Again, we observed that PRC2 localization to Xist domains (core PRC2 subunit Eed) is largely abolished in Jarid2 null female embryos (Supplementary Figs 4c, 5a,b,c). H3K27me3 deposition at Xist RNA domains was also

strongly reduced although not entirely abolished (Supplementary Figs 4b,5b,c), similar to the BglXist1 mESC model. Together, these findings suggest that Jarid2 plays a pivotal role in recruiting PRC2 to Xist-dependent H2AK119u1 domains. However, the presence of residual PRC2 recruitment following deletion of Jarid2 points to the existence of a parallel mechanism through which PRC2 can recognize H2AK119u1.

**A ubiquitin interaction motif (UIM) in Jarid2.** We went on to define which regions of the Jarid2 protein are required for recognition of H2AK119u1. Thus, we established Jarid2 null mESC sublines stably expressing HA-tagged full length, mutant or truncated Jarid2 proteins (Fig. 3a and Supplementary Fig. 6a). Consistent with previous reports[21], HA-tagged Jarid2 proteins, with the exception of the carboxy-terminal fragment (aa 542–1234), all interact with endogenous PRC2 (Supplementary Figs 6b,c, 7a,b). We then transfected the mESC sublines with the MBD-

RPCD construct, and monitored H3K27me3 deposition at PCH. The data are summarized in Fig. 3b, with examples shown in Supplementary Fig. 6d,e. Full length Jarid2 rescued H3K27me3 deposition at PCH, consistent with expectations, and this was unaffected by mutating Jarid2 lysine 116 (K116A), a residue that when methylated by PRC2 stimulates PRC2 activity both *in vitro* and *in vivo*[22] (Fig. 3a,b and Supplementary Fig. 6d,e). Interestingly, N-terminal fragments encompassing the trans-repression domain (TRD)/PRC2 interaction domain[21], and either with or without the JmjN domain, also fully complemented H3K27me3 deposition (Fig. 3a,b and Supplementary Fig. 6d,e). Conversely, expression of the C-terminal fragment encompassing JmjN, ARID, JmjC and ZF domains failed to restore H3K27me3 (Fig. 3a,b and Supplementary Fig. 6d,e). These data indicate that the Jarid2 N-terminal region preceding the JmjN domain harbours the key residues required for recognition of H2AK119u1-modified chromatin.

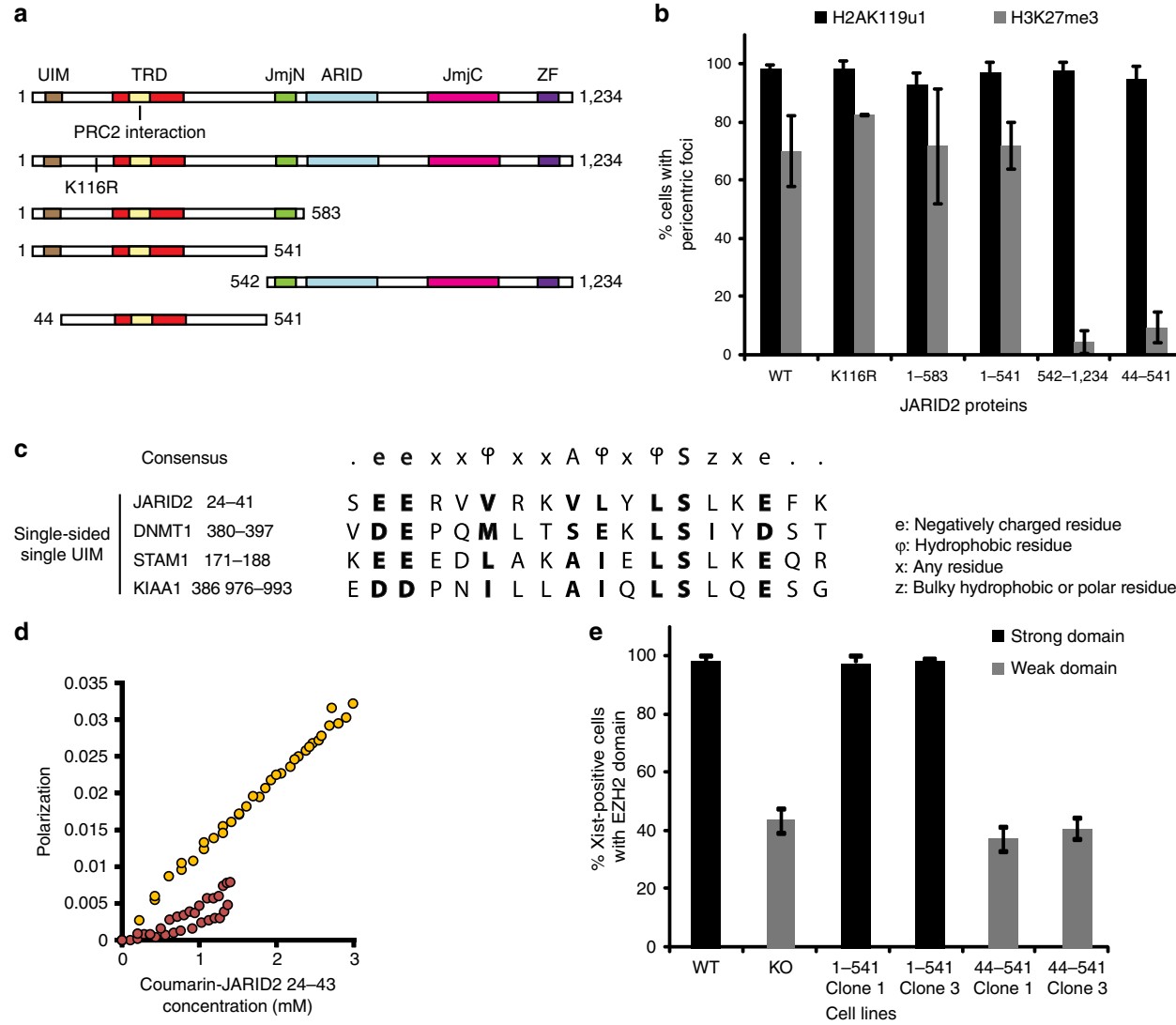

**Figure 3 | A ubiquitin binding motif in the Jarid2 N-terminus mediates interaction with H2AK119u1.** (**a**) Schematic showing full length JARID2 and complementation constructs, UIM, ubiquitin interaction motif; TRD, transcription repressive domain; JmjN, JumonjiN; JmjC, JumonjiC; and ZF, zinc finger domain. (**b**) Levels of restoration of H3K27me3 in response to H2AK119u1 after complementation by the constructs in **a**. (**c**) Consensus UIM sequence and alignment of single-sided UIMs with sequence from Jarid2 N-terminus. (**d**) JARID2 UIM peptide interacts with ubiquitin. Coumarin-JARID2 24–43 binds to WT ubiquitin (yellow) more strongly that I44A ubiquitin (red) as monitored by fluorescence polarization (*n* = 3). (**e**) Co-localization of EZH2 with Xist-Bgl-mCherry domains in wild type, Jarid2 null mESCs and in mESC cell lines complemented with the indicated constructs. A minimum of 200 Xist-Bgl-mCherry-positive cells were counted in three biological repeats in all experiments. Error bars indicate s.d.

A careful examination of the Jarid2 N-terminal region revealed a putative single-sided single UIM[23] at the extreme N-terminus, aa24–40 with close similarity to a UIM in Dnmt1 that interacts with ubiquitylated histone H3 (ref. 24) (Fig. 3c). The putative UIM is highly conserved in different vertebrate species (Supplementary Fig. 8a). UIMs of this class have been shown to form a short helix, which interacts with the I44 hydrophobic patch on ubiquitin[25]. Accordingly, using fluorescence polarization, we demonstrated that a Jarid2 UIM peptide binds to WT but not to I44A mutant ubiquitin (Fig. 3d). Although weak (>1 mM), this interaction is in the same range seen for other UIMs[25]. We went on to determine the importance of the UIM for PRC2 recruitment to PCH domains. As shown in Fig. 3a,b and Supplementary Fig. 6d,e, recognition of H2AK119u1 at PCH domains by the Jarid2 N-terminal region was largely abolished in the absence of the UIM (44–541 Jarid2 construct). Importantly, this mutation had no effect on the ability of the Jarid2 N-terminus to interact with PRC2 (Supplementary Figs 6c,7b). We further analysed the importance of the UIM using the inducible Xist transgene model described above. Complementation of Jarid2 null BglXist1 mESCs with wild-type Jarid2 N-terminus fully restored Xist-dependent recruitment of PRC2 (EZH2), consistent with a previous report[17], but this effect was entirely lost following deletion of the UIM (Fig. 3e and Supplementary Fig. 6f).

**The Jarid2 UIM directs interaction with H2AK119u1.** The aforementioned genetic analyses suggest that the Jarid2 N-terminus may bind directly to H2AK119u1-modified chromatin. To test this idea, we developed a pull-down assay using reconstituted unmodified or H2AK119u1 nucleosomes (Supplementary Fig. 8b), and nuclear extract from wild-type and Jarid2 null mESCs. RYBP, a core component of non-canonical PRC1 previously shown to bind to H2AK119u1 (ref. 26), was used as a control. As shown in Fig. 4a (Supplementary Fig. 9), in nuclear extract from wild-type mESCs, both Jarid2 and PRC2 (EZH2 subunit) bound strongly to H2AK119u1 compared with unmodified nucleosomes. Introduction of the I44A mutation in ubiquitin abolished the enhanced binding of both Jarid2 and PRC2, confirming that recognition of ubiquitin is important for the observed interaction. Enhanced binding of PRC2 was not seen using nuclear extract from Jarid2 null mESCs (Fig. 4a, right panel; and Supplementary Fig. 9).

To further substantiate direct interaction of the Jarid2 N-terminus and H2AK119u1, we expressed the N-terminal region of Jarid2, residues 1–530, as a recombinant protein (Supplementary Fig. 8c), and then assayed binding to either unmodified or H2AK119u1 mononucleosomes using biolayer interferometry (Fig. 4b). The affinity of the Jarid2 1-530 for H2AK119u1 was significantly higher (2.8 ± 0.5 μM), than that between ubiquitin and the Jarid2 UIM (>1mM), and showed a reproducible approximately twofold increase in affinity compared with unmodified nucleosomes (or H2AK119u1I44A) nucleosomes (5.0 ± 0.7 and 5.1 ± 1 μM, respectively). This difference was largely attributable to the $k_{on}$ being significantly faster (576 s$^{-1}$ for H2AK119u1 compared with 313 and 346 s$^{-1}$ for unmodified and H2AK119u1I44A, respectively). The binding of Jarid2 1–530 to unmodified and H2AK119u1 I44A nucleosomes is likely the result of a previously defined nucleosome interaction domain located between Jarid2 residues 349–450 (ref. 27).

**Discussion**

Previous studies have determined that Jarid2 plays an important role in the recruitment of PRC2 to target loci in ES cells[15,16,21,28,29] and the inactive X chromosome in female mammals[17]. Consistent with this, Jarid2 loss of function in mouse results in mid-late-gestation lethality[30]. In light of our findings, we suggest that these phenotypes are attributable to decreased interaction between PRC2 and H2AK119u1, affecting initiation and/or maintenance of Polycomb domains at target loci. The fact that embryo lethality occurs later in Jarid2 null mice[30], than with null mutations in genes encoding core PRC2 proteins, for example Eed[31], implies either H2AK119u1-independent targeting of PRC2 by sequence-specific DNA binding factors/lncRNAs, or alternatively, compensatory mechanism(s) for the recognition of H2AK119u1 by PRC2. In light of our finding that H2AK119u1 is required to initiate PRC2 recruitment by Xist RNA (unpublished), the observation that PRC2 recruitment to Xi is retained, at least to a small degree, in Jarid2 null ES cells, supports the latter possibility, although does not rule out the former.

The chromatin remodeller ATRX was previously reported to be highly enriched on Xi[32] and to play a key role in PRC2 recruitment by Xist RNA[19]. We on the other hand found that CRISPR/Cas9 mediated deletion of the Atrx gene had no effect on

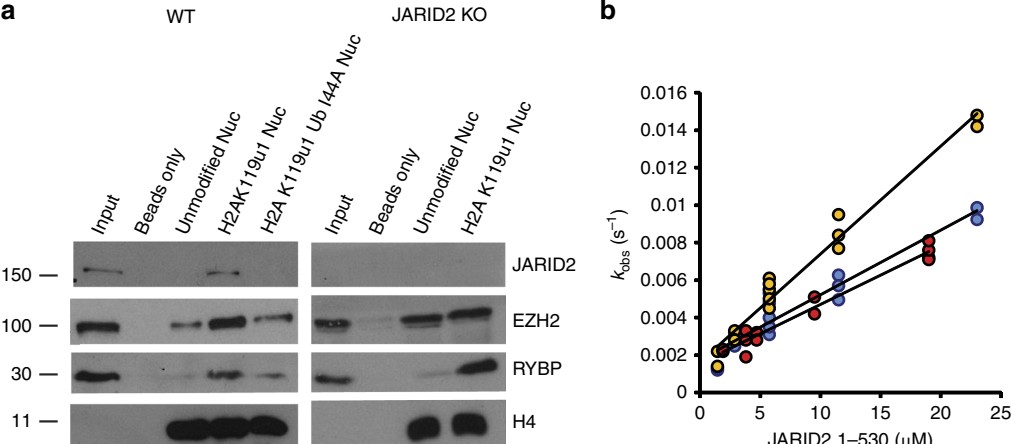

**Figure 4 | JARID2 N-terminus mediates direct interaction with H2AK119u1-modified nucleosomes** (**a**) Immunoblot of nucleosomal pull downs from nuclear extract. Total nuclear extract (input) and pull down using streptavidin beads, unmodified nucleosomes or nucleosomes modified by H2AK119u1/ H2AK119u1(I44A) were probed with the indicated antibodies using extracts from WT cells (left) or JARID2 KO mESCs (right). Full blots are shown in Supplementary Fig. 9. (**b**) JARID2 1–530 binds to H2AK119u1-modified mononucleosomes (yellow) more tightly than unmodified (blue) or H2AK199u1(I44A) (red) as monitored by biolayer interferometry (n = 3).

Xist-mediated recruitment of PRC2. Consistent with this finding, we have also been unable to detect ATRX enrichment on Xi in differentiating XX ES cells (unpublished). Our findings therefore indicate that prior evidence for the involvement of ATRX in PRC2 recruitment by Xist RNA needs to be reappraised.

In experiments performed using recombinant nucleosomes, we demonstrate that Jarid2 directly binds to H2AK119u1, and that this interaction involves a UIM in the Jarid2 N-terminus binding to the I44 patch of ubiquitin. While binding of the UIM to ubiquitin is relatively weak, we observed significantly enhanced binding using a recombinant Jarid2 N-terminal fragment. This difference is likely attributable to additional sequences in the Jarid2 N-terminus that enhance nucleosome binding[27]. It should be noted that the enhanced binding of the Jarid2 N-terminus to H2AK119u1 compared with unmodified nucleosomes is relatively modest, arguably insufficient to account for the enhanced activity of PRC2 on H2AK119u1 nucleosomes in vivo in itself. One possible explanation is that interaction surfaces contributed by other PRC2 subunits, or by the Jarid2 C-terminal domains, further enhances binding to H2AK119u1. An alternative, although not mutually exclusive idea, is that enhanced H3K27me3 deposition in vivo also reflects an allosteric switch in PRC2 activity triggered by Jarid2 binding to H2AK119u1. Consistent with this proposal, recent structural studies have defined a key allosteric mechanism for activation of PRC2 by ligands that bind to the aromatic cage of the EED subunit[33–35].

In summary, we define a key molecular mechanism by which the PRC2 cofactor Jarid2 directly binds to H2AK119u1-modified chromatin, providing insight into how PRC1 and PRC2 collaborate to initiate the formation of repressive chromatin at target genes through development and cellular differentiation.

## Methods

**Cell growth.** ES cells were grown in ES media (Dulbecco's Modified Eagle Medium (DMEM, Life Technologies) supplemented with 10% foetal calf serum (FCS, Seralab), 2 mM L-glutamine, 1× non-essential amino acids, 50 μM 2-mercaptoethanol, 50 μg ml$^{-1}$ penicillin/streptomycin (Invitrogen) and LIF-conditioned medium, made in house, at a concentration equivalent to 1000 U ml$^{-1}$. ESCs were grown on inactivated SNLP feeders (STO mouse fibroblasts expressing neomycin, puromycin resistance and Lif genes), unless otherwise stated. ES cell knockout (KO) medium was KnockOut DMEM (Life Technologies) with the same supplements as ES cell medium, plus 3.3% Knockout Serum Replacement (KnockOut SR Life Technologies). Cell lines used in this study were 129/1 WT and Aebp2$^{tr/tr}$ cells[36], Jarid2 WT and KO[15], Dnmt1$^{f/f}$ (ref. 13), BglXist1 (ref. 18). Jarid2 WT and KO cells were grown in ES cell KO medium without feeders, and Eed WT and KO cells were grown in ES medium without feeders.

**Transfection.** Transient and stable mESC lines were generated by transfection of 4 μg expression constructs using Lipofectamine 2000 (Life Technologies). Stable integrants were selected using 1.5 μg ml$^{-1}$ puromycin and individual colonies were picked and tested for expression of the construct by immunoblotting.

**CRISPR knockout cell lines.** Cells were transfected with the pX459 vector containing sgRNA target site sequences listed in Supplementary Table 1. Thirty hour after transfection, medium containing 1.5 μg ml$^{-1}$ puromycin was added, and after a subsequent 48 h, cells were cultured in medium without selection until colonies had grown. Successful homozygous mutation was confirmed by PCR and sequencing using primers in Supplementary Table 1 and, where possible by immunoblotting. Antibodies used were anti-JARID2 (Novus Biologicals NB100-2214) at 1:1,000 dilution, anti-SUZ12 (Cell Signalling 3737) at 1:1,000 dilution, anti-EZH2 (Cell Signalling 5246) at 1:1,000 dilution, anti-ATRX (a gift from R. Gibbons) at 1:10 dilution, anti-EED (a gift of A. Otte) at 1:500 dilution. Controls used were H2AK119u1 (NEB 8240) at 1:1,000 dilution, anti-H3K27me3 (Diagenode pAb-069-050) at 1:1,000 dilution, anti-H3 (Abcam ab1791) at 1:10,000 dilution and anti-RING1B (a gift of H. Koseki) at 1:1,000 dilution.

**Immunofluorescence.** ESCs were grown on slides without feeders, fixed with 2% formaldehyde for 15 min, permeabilized with 0.4% Triton X-100 for 5 min and blocked with 0.2% fish gelatin (Sigma) for 30 min. Slides were incubated with primary antibody for 2 h (diluted in 0.2% fish gelatin and 5% normal goat serum), washed three times and incubated with Alexa-fluor conjugated secondary antibody

for 2 h (Life Technologies). After washing five times, the slides were stained with DAPI (1 μg ml$^{-1}$), and mounted using mounting media (Dako). Primary antibodies used were protein-A purified anti-AEBP2 (ref. 36) at 1:10 dilution, anti-H3K27me3 (Active motif 39157) at 1:500 dilution, anti-SUZ12 (Cell Signalling 3737) at 1:500 dilution, anti-EZH2 (Cell Signalling 5246) at 1:500 dilution, anti-EED (a kind gift from A. Otte) at 1:100 dilution, anti-JARID2 (Novus Biologicals NB100-2214) at 1:500 dilution, H2AK119u1 (NEB 8240) at 1:500 dilution, anti-FLAG (Sigma M2 F1804 and F7425) at 1:500 dilution, anti-mCherry (Source Bioscience ABE3523 and gift from F. Barr) at 1:500 and 1:800 dilution, anti-RING1B at 1:500 dilution and anti-ATRX (a gift from R. Gibbons) at 1:10 dilution.

**Pericentric heterochromatin targeting assay.** Methyl-binding domain (MBD) fusion proteins were as described[13]. The RING1B/PCGF4 catalytic domain (RPCD) fusion was produced by linking amino acids 1–116 of mouse RING1B and 3–109 of PCGF4 using a 4× GGS flexible linker to create a minimal E3 ubiquitin ligase[13]. The MBD domain of human MBD1 (residues 1–112) was fused with a 2× GGS flexible linker to the N-terminus of RPCD or full length mouse Ezh2 along with a C-terminal SV40 NLS and FLAG-tag. These MBD-fusion proteins were cloned into the pCAG plasmid. Isogenic WT and mutant ESC lines used were 129/1 and Aebp2$^{tr/tr}$ (ref. 36) and Jarid2 WT and KO[15]. mESCs were transiently transfected using lipofectamine 2000 (Invitrogen), and fixed and stained 3 days post transfection for FLAG, H2AK119u1 and H3K27me3.

The JARID2 fragments used to complement the mutant phenotype were generated using primers listed in Supplementary Table 1. PCR products were digested with Sal I and Not I and cloned into the pCAGIPuro vector, in which expression of the insert is controlled by the constitutive β-actin promoter. All constructs contained a C-terminal HA tag for detection, and were stably transfected into Jarid2 KO mESCs.

Quantification of localization to PCH was carried out by counting the number of cells containing visible H2AK119u1 or H3K27me3-stained PCH foci compared with the total number of transfected cells (n > 300). Error bars show standard deviation of three biological repeats. All quantifications were carried out blind, with coded genotypes and represent the average counts obtained independent by two individuals.

**Redistribution of Polycomb in Dmnt mutant cells.** Conditional Dnmt1$^{f/f}$ ESCs[37] were grown on inactivated PEFS and gene deletion was induced by the addition of 800 nM 4-hydroxytamoxifen. Knockout of the Jarid2 gene was performed in this background using CRISPR/Cas9, and three independent mutant lines were generated. These were validated on the basis of a shift in the reading frame of the protein determined by Sanger sequencing, and the absence of detectable JARID2 protein by immunoblotting.

Dnmt1$^{f/f}$ and three Dnmt1$^{f/f}$Jarid2$^{-/-}$ knockout lines were grown for 6 and 12 days with or without 4-hydroxytamoxifen addition, and immunostaining was performed for H2AK119u1 and H3K27me3. The number of cells with PCH foci positive for H2AK119u1 or H3K27me3 were counted as a percentage of the total number of cells (n > 300). Three biological repeats were performed for each of the three independent mutant lines and controls. Error bars show standard deviation of the three mutant lines, and counting was carried out blind as described above.

C-terminally eGFP tagged EZH2 was cloned into the pCAGIPuro vector, and stable cell lines were generated in the Dmnt1$^{f/f}$ and Dmnt1$^{f/f}$ Jarid2$^{-/-}$ backgrounds. Clones expressing a comparable level of tagged EZH2 to the endogenous protein, as determined by immunoblotting, were selected for further experiments. Live imaging was performed at 0 and 10 days following 4-hydroxytamoxifen addition and the number of cells with PCH foci positive for EZH2-eGFP were counted as a percentage of the total number of cells (n > 300) as above.

**Chromatin targeting by Xist.** Full length Xist cDNA was cloned into the pTRE-tight vector with Bgl stem loops inserted within the sequence to allow visualization of Xist localization in live cells, and a tetO promoter driving its expression[18]. Male ESCs, that constitutively expressed the tetracycline transactivator protein, rtTA such that expression of Xist can be regulated by addition of doxycycline were co-transformed with the Xist-Bgl construct and a construct expressing an mCherry tagged BglG protein to label the Xist domains, and stable lines were generated. Clones were selected whereby induction of Xist induced a single domain of Xist localization, due to integration of the Xist transgene on a single autosome[18].

Knockout of the Jarid2, Atrx and Suz12 genes were performed in this background using CRISPR/Cas9 as above. Mutant clones were validated on the basis of a shift of the reading frame of the protein determined by Sanger sequencing, and absence of detectable JARID2, ATRX and SUZ12 proteins using immunoblotting.

Expression of the Xist transgene was induced with 1.5 ug ml$^{-1}$ doxycycline for 1 day in WT, Jarid2$^{-/-}$, Atrx$^{-/-}$ and Suz12$^{-/-}$ cells. Immunostaining was performed for H3K27me3, EZH2, SUZ12 and mCherry. The number of cells with an inactive chromosome focus stained with these markers were counted as a percentage of the total number of cells (n > 300). Three biological repeats were performed for each of the mutant lines and controls. Error bars show standard

deviation of the three biological repeats, and counting was carried out as described above.

**Co-immunoprecipitation of endogenous and HA-tagged proteins.** Cells from a confluent 140 mm plate were collected and resuspended in 500 μl cell lysis buffer C300 (20 mM HEPES-KOH pH 7.9, 1.5 mM MgCl₂, 0.1 % NP40, 0.2 mM EDTA-NaOH pH8.0, 300 mM KCl with freshly added 0.5 mM DTT, cOmplete EDTA-free (Roche)). Cells were incubated on ice for 20 min and cell debris was removed by centrifugation (16,100$g$, 20 min, 4 °C). Extracts were pre-cleared with 30 μl packed volume salmon sperm-blocked Protein A Agarose beads (Millipore) in the presence of 250 U Benzonase Nuclease (Millipore) for 30 min at 4 °C. Antibodies were added (anti-EZH2 (Cell Signalling 5246, 5 μl), IgG (Sigma M7023, 4 μl) and HA affinity matrix (Roche 3F10, 11867423001, 15 μl)) and incubated at 4 °C overnight. Subsequently, 30 μl packed volume Protein A Agarose beads were added to capture antibodies against endogenous proteins and incubated for 2 h at 4 °C. The flow through was collected and beads were washed five times with 1 ml C300. Subsequently beads were boiled in 50 μl SMASH buffer (50mM Tris-HCl pH 6.8, 10% Glycerol, 2% SDS, 0.02% bromophenol blue, 1% beta-mercaptoethanol) for 5 min at 95 °C. Immunoblots were probed with anti-HA (Roche 3F10) at 1:1,000 dilution, anti-H3 (Abcam ab1791) at 1:10,000 dilution and anti-EZH2 (Cell Signalling 5246) at 1:1,000 dilution.

**Southern blotting.** For methylation-specific southern blots, genomic DNA was digested with the methylation-sensitive enzyme HpyCH41V, separated on a 1.25% agarose gel and blotted to a hybond-XLMembrane. Blots were probed with a 'gamma' mouse major satellite probe as described[38,39] labelled using klenow and alpha-P-32-ATP.

**Genetic crosses.** All animals used in the studies were handled with care and experiments were done according to the guidelines from the Danish animal ethical committee (Dyreforsøgstilsynet) and French legislation and institutional policies. Targeted Jarid2 mice (C57Bl/6N.Jarid2^tm1a(KOMP)Wtsi) were obtained from the Knockout Mouse Program (KOMP, https://www.mousephenotype.org/data/genes/MGI:104813). In these animals, a lacZ-Neo-reporter cassette flanked by FRT sites is inserted between exons 1 and 2 of Jarid2 as well as LoxP sites surrounding exon 2. To obtain conditional (Jarid2^flox/flox) and Jarid2 knockout (Jarid2^ko) alleles, they were sequentially crossed with mice ubiquitously expressing Flp- and Cre-recombinase. Maternal and paternal preimplantation Jarid2-deficient embryos were generated from natural mating between Jarid2^flox/flox;Zp3::cre^+/− females with Jarid2^ko/+ or Jarid2^ko/flox males. Animals were maintained on a C57Bl/6 background and the different Jarid2 alleles (wt 994 bp, Neo 719 bp, flox 1192 bp, ko 240 bp) were detected with primers listed in Supplementary Table 1.

**Embryo collection and immunostaining.** Blastocysts were collected in M2 medium (Sigma) by flushing the uterus. The zona pellucida was removed, embryos were transferred onto coverslips previously coated in Denhardt's solution and dried for 30 min at room temperature as described[40]. Samples were fixed in 3% paraformaldehyde (pH 7.2) for 10 min at RT and permeabilized in ice-cold PBS, 0.5% Triton X-100 containing 2 mM Vanadyl Ribonucleoside Complex (NEB) on ice for 3–15 min. After rinsing with PBS, embryos were blocked with PBS 1% BSA (Gibco 15260-037) and 1 U μl RNAse inhibitor (Euromedex) for 15 min then incubated with primary antibodies (Jarid2 (ab48137; 1:800), Eed (A.Otte 1:100) and H3K27me3 (Active Motif 39155 1:200)) diluted in PBS 1% BSA, 4 U μl RNAse inhibitors for 1.5 h. Coverslips were then washed three times in PBS and incubated with secondary antibody (1:250) for 30 min. After washing, preparations were postfixed in 4% PFA for 10 min and rinsed in SSC 2 ×. For RNA FISH, the Xist probe was labelled with Spectrum Red-dUTP (Vysis) by nick translation on a preparation of a plasmid containing a 19 kb genomic fragment covering most of the Xist gene. Hybridizations using 0.1 μg of probe per coverslip in 50% formamide, 2 × SSC, 20% dextran sulfate, 1 mg ml⁻¹ BSA (Biolabs) and 200 mM VRC were performed overnight at 37 °C in a humid chamber followed by three washes in 2 × SSC at 42 °C for 5 min. Slides were mounted in Vectashield (Cliniscience) containing DAPI. Images were acquired on a Zeiss LSM700 inverted confocal microscope with a Plan apo DICII (numerical aperture 1.4) × 63 oil objective. Z sections were taken every 1 μm and full Z stack projections are shown. Images were analysed using ImageJ software.

**Nuclear extract preparation.** Cells were collected, washed in PBS and resuspended in 10 packed cell volumes buffer A (10 mM HEPES, pH 7.9, 1.5 mM MgCl₂, 10 mM KCl, 0.5 mM DTT, 0.5 mM PMSF, protease inhibitors (Roche)). Cells were collected and resuspended in three cell volumes of buffer A containing 0.1% NP-40, and incubated on ice for 10 min. Nuclei were collected and resuspended in one cell volume buffer C (5 mM HEPES, pH 7.9, 26% glycerol, 1.5 mM MgCl₂, 0.2 mM EDTA, 250 mM NaCl, 0.5 mM DTT, protease inhibitors), before NaCl concentration was increased to 350 mM by adding 5 M NaCl dropwise while mixing. Salt extraction was carried out for 1 h on ice, and chromatin pelleted at 16,000$g$ for 20 min at 4 °C. The supernatant was taken as soluble nuclear extract, and quantified by Bradford assay.

**Histone purification and nucleosome reconstitution.** Recombinant Xenopus histones were expressed in bacteria and purified from inclusion bodies. Stoichiometric amounts of each core histone were incubated together under high-salt conditions, and the resulting histone octamer was purified using a Superdex 200 gel filtration column (GE Healthcare). His-tagged ubiquitin was cloned into a PET14b plasmid, expressed in BL21 cells, sonicated, clarified by centrifugation and purified by affinity purification. Site-directed mutagenesis was performed using the QuikChange Lightning kit (Aligent). Ubiquitinated H2A was produced as previously described[41]. In brief, ubiquitin, mutated to include a C-terminal Cysteine (G67C), was crosslinked to H2A K119C using 1,3-dichloroacetone. Cross linked product was purified by affinity purification via the N-terminally tagged ubiquitin.

For pull downs, biotinylated DNA containing two repeats of the nucleosome positioning sequence (601) and 48 bp linker was amplified by PCR (5′ biotin-tagged primer) and purified. For biolayer interferometry, a biotinylated 147 bp DNA fragment containing a single 601 sequence was produced as described[42].

Equimolar ratios of DNA and octamers were mixed together in 2 M NaCl and diluted stepwise with 10 mM Tris-HCl, pH 7.5, to reach a final concentration of 100 mM NaCl. The reconstituted dinucleosomes were analysed by an electrophoretic mobility shift assay (EMSA) using 0.8% agarose gel in 0.2% Tris-borate and post stained with ethidium bromide.

**H2AK119u1 pull downs.** WT and uH2A dinucleosomes were incubated with prewashed, magnetic, streptavidin coated beads (Fisher) for 1 h at 4 °C, and then washed five times with 250 mM NaCl, 10 mM Tris-HCl pH 7.0 and 0.01% Tween 20. 10 μg dinucleosomes (WT or uH2A) were incubated with 250 ug nuclear cell extract for 1 h at 4 °C, again washed five times with 250 mM NaCl, 10 mM Tris-HCl pH 7.0 and 0.01% Tween 20, and finally the reaction stopped by adding SDS loading buffer. Samples were analysed by SDS-PAGE and western blot. Primary antibodies used were anti-HA (Roche 3F10 11867423001) at 1:1,000 dilution, anti-ATRX (a gift from R.Gibbons) at 1:10 dilution, anti-EZH2 (Cell Signalling 5246) at 1:1,000 dilution, anti-JARID2 (Novus Biologicals NB100-2214) at 1:1,000 dilution and anti-H4 (Abcam AB7311).

**JARID2 purification.** hJARID2 1–530 virus was a gift from Raphael Margueron, cloned using the Bac-to-Bac Baculovirus Expression Systems (Invitrogen). Protein was produced in SF9 insect cells after infection. Lysates containing Flag-tagged proteins were resuspended in 300 mM NaCl, 20 mM Tris-HCl, pH 8.0, 2.5 mM dithiothreitol (DTT), lysed by sonication and clarified by centrifugation. Following incubation with Flag beads (M2-beads), protein was eluted by Flag peptide overnight.

**Biolayer interferometry.** Jarid2 1–530 binding to immobilized mononucleosomes was measured on an Octet RED biolayer interferometer (Pall ForteBio Corp., Menlo Park, CA, USA). Biotinylated mononucleosomes were immobilized on streptavidin biosensors (Pall ForteBio Corp., Menlo Park, CA, USA) at a concentration of ∼0.012 μg ml⁻¹. The binding of Jarid2 (at 2–25 μM) to the immobilized nucleosomes was measured at 25 °C with a 500 s association step followed by a 1,000 s dissociation step. The buffer was 10 mM HEPES (pH 7.4), 150 mM NaCl, 3 mM EDTA, 0.005% Tween-20 and 1 mg ml⁻¹ BSA. The association phase was analysed as a single exponential function using in-house software and a plot of the observed rate ($k_{obs}$) versus the Jarid2 concentration gave the association and dissociation rate constants ($k_{on}$ and $k_{off}$) as the slope and intercept, respectively, (Fig. 4b). The dissociation rate constant was also determined independently from analysis of the dissociation phase.

**Fluorescence polarization.** WT or I44A ubiquitin was titrated into 7 μM fluorescent UIM peptide (7-Methoxycoumarin-4-Acetic acid –SEERVVRKVLYLSLK-EFKNA) purchased from GL Biochem Ltd. Fluorescence anisotropy was monitored on a JASCO FP 8500 in 100 mM NaCl and 50 mM Tris pH 8.0, data was analysed using in-house software.

**Data availability.** The data that support the findings of this study are available from the corresponding author upon request.

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

## Acknowledgements
We thank the members of the Brockdorff and Klose labs for helpful discussions and critical comments on the manuscript, Arie Otte, Francis Barr, Richard Gibbons and Haruhiko Koseki for antibodies and cell lines and Raphael Margueron and Till Bartke for expression constructs. The Fluorescence Polarization and biolayer Interferometry experiments were performed at The Structural Biology Science Technology Platform, The Francis Crick Institute, London, and we thank Stephen Martin for guidance in these experiments. Work in N.B. lab was funded by grants from the European Research Council (340081) and Wellcome Trust (103768). Work in K.H. lab is supported by the European Research Council (294666). Danish Cancer Society, the Danish National Research Foundation (DNRF 82), the Lundbeck Foundation and through a centre grant from the Novo Nordisk Foundation (The Novo Nordisk Foundation Section for Stem Cell Biology in Human Disease). S.M.K. was supported by a postdoctoral fellowship from the Netherlands Organization for Scientific Research (NWO). E.H. acknowledges the pathogen-free barrier animal facility of the Institut Curie, in particular Colin Jouhanneau, and the UMR3215/U934 Imaging Platform (PICT-IBiSA), in particular Olivier Leroy and Nicolas Signolle. Funding for E.H.: Labex DEEP (ANR-11-LBX-0044) part of the IDEX Idex PSL (ANR-10-IDEX-0001-02 PSL) and ERC Advanced Investigator award (250367).

## Author contributions
S.C., E.U., K.H., E.H. and N.B. conceptualized the experiments and supervised the study. S.C., A.G., E.U., K.A., T.Z., T.B.N., B.A.-K., A.B., S.M.K. and K.A. performed the experiments. S.C., E.U., T.B.N. and N.B. prepared the manuscript. The manuscript has been approved by all authors. N.B. secured the funding.

## Additional information

**Competing financial interests**: The authors declare no competing financial interests.

**How to cite this article**: Cooper, S. *et al.* Jarid2 binds mono-ubiquitylated H2A lysine 119 to mediate crosstalk between Polycomb complexes PRC1 and PRC2. *Nat. Commun.* **7**, 13661 doi: 10.1038/ncomms13661 (2016).

**Publisher's note**: 

