## [Peer Review File · Nature Communications]

PEER REVIEW FILE

Reviewers' comments:

Reviewer #1 (Remarks to the Author):

Cooper et al present a new version of their study on the interaction of PRC2 with monoubiquitylated H2A. The issue with the previous version of the paper was that many of the data were not novel and that there were serious problems with biochemical data that were central to the manuscript. In this version of the manuscript, the authors removed most of those problematic data. They include new data. Specifically, the authors identify an ubiquitin interaction motif (UIM) in the N-term of Jarid2 and the authors present data that suggest that this motif binds to ubiquitin. The affinity of this interaction is very low (> 1mM). The authors also show that a larger portion, comprising the N-term, Jarid2[1-530], binds slightly better to nucleosomes with monoubiquitylated H2A compared to unmodified H2A. From these data, the authors conclude that “Direct interaction of Jarid2 with mono-ubiquitylated H2A lysine 119 mediates crosstalk between Polycomb complexes PRC1 and PRC2” (title of the manuscript).

I still think that this is a very minor advance compared to what was published in previous studies and the newly added data again have several technical issues. Considering that the affinity of the Jarid2 UIM interaction with ubiquitin is so low, it is hard to believe that simple disruption of Jarid2 with H2Aub nucleosomes could account for the IF data in Fig. 3b, e and in the extended data Fig. 6. The spliced images of westerns with cropped single bands, purporting to show that the N-terminally truncated Jarid2 protein is present and interacts with E(z) like the wild-type protein does not help here (Extended data Fig. 6); these bands could be anything and from totally different exposures. In this form, those data are certainly unacceptable for publication. Also, there are no data in this study that document that point mutation of the Jarid2 UIM in the context of a PRC2 complex containing Jarid2 and Aebp2 would abolish the preferential binding of this complex to nucleosomes with monoubiquitylated H2A or reduce H3K27 methylase activity on these nucleosomes in vitro. Given these shortcomings and the lack of novelty and advance, I am not able to support publication of this work in Nature Communications.

Reviewer #3 (Remarks to the Author):

The authors have adequately responded to my previous issues so that the manuscript is ready to be published from my point of view.

Reviewer #4 (Remarks to the Author):

It has been previously demonstrated that the recruitment of PRC2 to chromatin lead to the recruitment of PRC1. Yet, for years the mechanism underlines the recruitment of PRC2 remained obscured. During this time, various observations that were made in multiple independent studies led to propose models that explained the recruitment of PRC2 to chromatin by interactions with large cohort of protein factors and long non-coding RNAs. In the meanwhile, JARID2 has demonstrated to play a key role in the recruitment of PRC2 to chromatin. More recently, PRC2 was shown to be recruited to chromatin following the recruitment of PRC1. Yet, there was still thick ambiguity around the way JARID2 identify polycomb target genes. It was also not clear what is the missing bridging factor between PRC2 and nucleosomes that were ubiquitylated by PRC1. Although one could speculate around models to seemingly address these questions, neither direct evidence nor validation were provided to explain these association events; so far. In this manuscript, Cooper et al. providing a new function for JARID2 in PRC1-dependent recruitment of PRC2 to chromatin. Specifically, JARID2 bridges between PRC2 to nucleosomes that were ubiquitylated by PRC1.

In their study, Cooper et al. utilized large variety of experimental systems to discover, demonstrate and validate the mechanism of JARID2-mediated recruitment of PRC2 to ubiquitylated nucleosomes. Using multiple JARID2 knockout embryonic cell lines, they identified JARID2 as an essential factor for efficient recruitment of PRC2 to previously deposited H2AK119ub mark. They further validated these observations using preimplantation mouse embryos carrying Jarid2 conditional knockout. Series of deletions allowed them to identify the N-terminal of JARID2 as the bridging component between PRC2 and the H2AK119ub mark. This observation was thoroughly validated through elegant rescue experiments, where this portion of JARID2 was overexpressed to restore PRC2-H2AK119ub interactions in JARID2 KO cells.

Through this manuscript Copper et al. are revealing an important piece of the polycomb puzzle which allows, for the first time, to fully describe how PRC1 and PRC2 are functioning together at the molecular level, for the formation of stably repressed polycomb target genes. This is an important and solid work that will likely draw large attention from broad audience and will have essential contribution to the field of polycomb. I believe this manuscript is well suitable for publication in Nature Communications.

Though, given the large revision that performed since the previous submission, there are few issues that I believe can be fixed without further delaying publication:

> Major concern:

Figure 4 and Extended Data Figure 7: The quality of the data in these figures is not well aligned with the overall high quality of data elsewhere in this manuscript. There is no evidence for the reconstitution of the nucleosomes carries mutant ubiquitin mark (i.e. no H2AK119u1I44A in Extended Data Figure 7). This is important, because the very small difference in affinity measured by the biolayer interferometry assay (Figure 4b, approximately 2-fold) could potentially be explained by variations in quality between nucleosome samples. The recombinant protein that used in these experiments, Jarid2 1-530, was expressed and purified from insect cells. Yet, there is no Coomassie stained SDS-PAGE to assess the amount of contaminants carried with the recombinant protein. Furthermore, it is impossible to exclude the possibility that this protein was obtained in the form of soluble aggregates, a common problem that takes place when truncated proteins are overexpressed in Sf9 cells, especially in the absence of binding partners. The formation of such soluble aggregates can usually be excluded by using gel filtration chromatography, but this wasn't done downstream the affinity chromatography (flag/M2). In case that Jarid2 1-530 appears in soluble aggregates, what would be the implications on the rate constants and dissociation constants that were calculated, based on the biolayer interferometry measurements? How this will affect the apparent affinity if multiple binding sites within multiple proteins that trapped together in the same aggregate are exposed to interactions or, contrarily, hindered within the aggregate? How results will be interpreted if it will later be evident that Jarid2 1-530 is bringing multiple binding partners from the Sf9 host cell (e.g. abundant chaperons), which could easily be detected on an SDS-PAGE by Coomassie stain, but not by the provided blot (Extended Data Figure 7)? Although having quantitative biophysical data is usually a good direction, the way these experiments performed spreads large ambiguity around these results and the extent they can firmly support the conclusions. Accordingly, I see a great value for this work as a whole, even without the inclusion of these two figures and the conclusions that were derived from them. But if the authors will choose to include such data after all (Figure 4 and Extended Data Figure 7), it would be good to highlight some of the limitations behind such unpolished assays.

> Specific points that can be improved:

1. Figure 3e and Extended Data Figure 6: While JARID 1-541 demonstrated impressive ability to rescue the JARID2 KO phenotype, JARID 44-541 fail to do so. The authors used this data to

state that the Jarid2 N-terminus may bind directly to H2AK119ub modified chromatin. Indeed, this negative result is in agreement with some of the in vitro analyses. It is also consistent with the similarity demonstrated between several ubiquitin interaction motif (UIM) sequences from other proteins (Figure 3c). Yet, deleting 43 residues from the N-terminal of a protein can perturb function in many different ways, some that are irrelevant to this story (e.g. poor solubility, unstable fold, wrong fold, acquiring irrelevant binding partners etc.). In order to strengthen their statement, it is recommended that the authors will at least demonstrate evolution conservation for the key amino acids ("consensus" in Figure 3c) within the UIM of various JARID2 sequences in metazoa.

2. Page 4, 2nd paragraph: "In recent work we have shown that H2AK119u1 mediated by a variant PRC1 complex initiates Xist RNA dependent Polycomb domain formation in X chromosome inactivation (unpublished)." -- > 'In recent work' is mutually exclusive with 'unpublished'. Unless the unpublished manuscript will be accepted for publication, by the time this manuscript will be published, referring to unpublished data is better avoided.

3. Extended Data Figure 6b-c: although cropped blots are usually not harmful, some of the blots in these two panels were cropped down to a single band, which is less common in publications at this level. It might be good to add a full membrane capture as a supplemental.

4. Page 7, 1st paragraph: "Moreover, the interaction with ubiquitylated substrate shows clear differences in its binding kinetics, with the K_{on} being significantly faster"  When referring to rate constants, it is common to use small k_{on} , not capitalized K_{on} . It would also be good if actual values will be provided. Additionally, it is not clear how this statement related to the sentence that comes just above it, which describes rather low difference in affinity (~2-fold in term of dissociation constants, K_d ?) between Jarid2 1-530 and the two types of ubiquitylated nucleosomes (wild type vs mutant).

Reviewers' comments:

Reviewer #1 (Remarks to the Author):

Cooper et al present a new version of their study on the interaction of PRC2 with monoubiquitylated H2A. The issue with the previous version of the paper was that many of the data were not novel and that there were serious problems with biochemical data that were central to the manuscript. In this version of the manuscript, the authors removed most of those problematic data. They include new data. Specifically, the authors identify an ubiquitin interaction motif (UIM) in the N-term of Jarid2 and the authors present data that suggest that this motif binds to ubiquitin. The affinity of this interaction is very low ($> 1\text{mM}$). The authors also show that a larger portion, comprising the N-term, Jarid2[1-530], binds slightly better to nucleosomes with monoubiquitylated H2A compared to unmodified H2A. From these data, the authors conclude that "Direct interaction of Jarid2 with mono-ubiquitylated H2A lysine 119 mediates crosstalk between Polycomb complexes PRC1 and PRC2" (title of the manuscript).

I still think that this is a very minor advance compared to what was published in previous studies and the newly added data again have several technical issues. Considering that the affinity of the Jarid2 UIM interaction with ubiquitin is so low, it is hard to believe that simple disruption of Jarid2 with H2Aub nucleosomes could account for the IF data in Fig. 3b, e and in the extended data Fig. 6. The spliced images of westerns with cropped single bands, purporting to show that the N-terminally truncated Jarid2 protein is present and interacts with E(z) like the wild-type protein does not help here (Extended data Fig. 6); these bands could be anything and from totally different exposures. In this form, those data are certainly unacceptable for publication. Also, there are no data in this study that document that point mutation of the Jarid2 UIM in the context of a PRC2 complex containing Jarid2 and Aebp2 would abolish the preferential binding of this complex to nucleosomes with monoubiquitylated H2A or reduce H3K27 methylase activity on these nucleosomes *in vitro*. Given these shortcomings and the lack of novelty and advance, I am not able to support publication of this work in Nature Communications.

In the original submission we commented on the relatively weak interaction of the UIM with ubiquitin, highlighting that this is typical for UIM/ubiquitin interactions in other proteins. The specificity and overall affinity thus comes from additional interacting surfaces, and in this respect we show that the Jarid2 N-terminus interacts with H2AK119ub1 nucleosomes with low micromolar affinity. In the revised manuscript we further comment on the possibility that other PRC2 subunits contribute to the overall affinity and/or Jarid2 binding to H2AK119ub1 functions by allosterically activating PRC2. For the colP data in Extended data Fig. 6 (now Supplementary Fig 6), we have now included scans of the entire western blots (Supplementary Fig 7). Regarding point mutations in the UIM, it is the case that we do not currently know of a single residue mutation that disrupts this interaction. However, we do show that a point mutation in the hydrophobic I44 patch of ubiquitin abolishes the interaction and believe that this is clear evidence of specificity. Structural analysis of the interaction between the UIM and ubiquitin may highlight key residues that could be mutated, but this is beyond the scope of the current study. Finally, regarding novelty, this work represents the first clear demonstration of a mechanism underlying recognition of H2AK119ub1 by PRC2. Moreover, we define for the first time a specific function to the PRC2 cofactor Jarid2, previously implicated in PRC2

recruitment in normal physiology and in disease.

Reviewer #3 (Remarks to the Author):

The authors have adequately responded to my previous issues so that the manuscript is ready to be published from my point of view.

We thank the reviewer for their previous input.

Reviewer #4 (Remarks to the Author):

It has been previously demonstrated that the recruitment of PRC2 to chromatin lead to the recruitment of PRC1. Yet, for years the mechanism underlines the recruitment of PRC2 remained obscured. During this time, various observations that were made in multiple independent studies led to propose models that explained the recruitment of PRC2 to chromatin by interactions with large cohort of protein factors and long non-coding RNAs. In the meanwhile, JARID2 has demonstrated to play a key role in the recruitment of PRC2 to chromatin. More recently, PRC2 was shown to be recruited to chromatin following the recruitment of PRC1. Yet, there was still thick ambiguity around the way JARID2 identify polycomb target genes. It was also not clear what is the missing bridging factor between PRC2 and nucleosomes that were ubiquitylated by PRC1. Although one could speculate around models to seemingly address these questions, neither direct evidence nor validation were provided to explain these association events; so far. In this manuscript, Cooper et al. providing a new function for JARID2 in PRC1-dependent recruitment of PRC2 to chromatin. Specifically, JARID2 bridges between PRC2 to nucleosomes that were ubiquitylated by PRC1.

In their study, Cooper et al. utilized large variety of experimental systems to discover, demonstrate and validate the mechanism of JARID2-mediated recruitment of PRC2 to ubiquitylated nucleosomes. Using multiple JARID2 knockout embryonic cell lines, they identified JARID2 as an essential factor for efficient recruitment of PRC2 to previously deposited H2AK119ub mark. They further validated these observations using preimplantation mouse embryos carrying Jarid2 conditional knockout. Series of deletions allowed them to identify the N-terminal of JARID2 as the bridging component between PRC2 and the H2AK119ub mark. This observation was thoroughly validated through elegant rescue experiments, where this portion of JARID2 was overexpressed to restore PRC2-H2AK119ub interactions in JARID2 KO cells.

Through this manuscript Copper et al. are revealing an important piece of the polycomb puzzle which allows, for the first time, to fully describe how PRC1 and PRC2 are functioning together at the molecular level, for the formation of stably repressed polycomb target genes. This is an important and solid work that will likely draw large attention from broad audience and will have essential contribution to the field of polycomb. I believe this manuscript is well suitable for publication in Nature Communications.

Though, given the large revision that performed since the previous submission, there are few issues that I believe can be fixed without further delaying publication:

> Major concern:

Figure 4 and Extended Data Figure 7: The quality of the data in these figures is not well aligned with the overall high quality of data elsewhere in this manuscript. There is no evidence for the reconstitution of the nucleosomes carries mutant ubiquitin mark (i.e. no H2AK119u1I44A in Extended Data Figure 7). This is important, because the very small difference in affinity measured by the biolayer interferometry assay (Figure 4b, approximately 2-fold) could potentially be explained by variations in quality between nucleosome samples. The recombinant protein that used in these experiments, Jarid2 1-530, was expressed and purified from insect cells. Yet, there is no Coomassie stained SDS-PAGE to assess the amount of contaminants carried with the recombinant protein. Furthermore, it is impossible to exclude the possibility that this protein was obtained in the form of soluble aggregates, a common problem that takes place when truncated proteins are overexpressed in Sf9 cells, especially in the absence of binding partners. The formation of such soluble aggregates can usually be excluded by using gel filtration chromatography, but this wasn't done downstream the affinity chromatography (flag/M2). In case that Jarid2 1-530 appears in soluble aggregates, what would be the implications on the rate constants and dissociation constants that were calculated, based on the biolayer interferometry measurements? How this will affect the apparent affinity if multiple binding sites within multiple proteins that trapped together in the same aggregate are exposed to interactions or, contrarily, hindered within the aggregate? How results will be interpreted if it will later be evident that Jarid2 1-530 is bringing multiple binding partners from the Sf9 host cell (e.g. abundant chaperons), which could easily be detected on an SDS-PAGE by Coomassie stain, but not by the provided blot (Extended Data Figure 7)? Although having quantitative biophysical data is usually a good direction, the way these experiments performed spreads large ambiguity around these results and the extent they can firmly support the conclusions. Accordingly, I see a great value for this work as a whole, even without the inclusion of these two figures and the conclusions that were derived from them. But if the authors will choose to include such data after all (Figure 4 and Extended Data Figure 7), it would be good to highlight some of the limitations behind such unpolished assays.

We thank the reviewer for their constructive suggestions. We concur that there are limitations inherent in the biochemical and biophysical assays and acknowledge this both herein and in the discussion of the revised manuscript. In our view a key control is the I44A ubiquitin mutant that clearly indicates that the observed binding/interaction is specific and not linked to protein aggregation etc. Regarding preparation of recombinant nucleosomes we routinely check the quality of histone octamers and nucleosome reconstitutions for initial and replicate experiments. We have now replaced the image in Extended data Figure 7A (now Supplementary Fig 8B) with a gel image illustrating the quality of the unmodified, H2AK119u1 and H2AK119I44Aub1 octamers. We analysed Jarid2 1-530 by western because the amount of protein available was relatively small. In general, expression of recombinant Jarid2 N-terminus is challenging due to intrinsic protein instability. The construct for this protein was obtained from Raphael Margueron at the Institut Curie. Raphaels lab have characterised the product extensively, both in terms of demonstrating homogeneity on coomassie stained gels, and also in terms of protein behaviour on gel filtration columns, which demonstrated absence of protein aggregates. Their work with this protein product is published in Son et al, Genes and Dev, 2014.

> Specific points that can be improved:

1. Figure 3e and Extended Data Figure 6: While JARID 1-541 demonstrated impressive ability to rescue the JARID2 KO phenotype, JARID 44-541 fail to do so. The authors used this data to state that the Jarid2 N-terminus may bind directly to H2AK119ub modified chromatin. Indeed, this negative result is in agreement with some of the in vitro analyses. It is also consistent with the similarity demonstrated between several ubiquitin interaction motif (UIM) sequences from other proteins (Figure 3c). Yet, deleting 43 residues from the N-terminal of a protein can perturb function in many different ways, some that are irrelevant to this story (e.g. poor solubility, unstable fold, wrong fold, acquiring irrelevant binding partners etc.). In order to strengthen their statement, it is recommended that the authors will at least demonstrate evolution conservation for the key amino acids ("consensus" in Figure 3c) within the UIM of various JARID2 sequences in metazoa.

We have now included an alignment of UIM sequences from several vertebrate species (Supplementary Fig 8A). The Drosophila homologue of Jarid2 does not have an obvious homologous sequence, despite the fact Drosophila PRC2 has also been shown to bind to H2AK119u1. At this time it is not known if this is because the equivalent domain is too degenerate to identify by homology searching or whether in Drosophila there are alternative mechanisms for H2AK119u1 recognition by PRC2.

2. Page 4, 2nd paragraph: "In recent work we have shown that H2AK119u1 mediated by a variant PRC1 complex initiates Xist RNA dependent Polycomb domain formation in X chromosome inactivation (unpublished)."  'In recent work' is mutually exclusive with 'unpublished'. Unless the unpublished manuscript will be accepted for publication, by the time this manuscript will be published, referring to unpublished data is better avoided.

We have reworded this sentence. Clearly it is necessary to refer to this data in order to make sense of the experiments investigating the role of Jarid2 in PRC2 recruitment by Xist RNA. A manuscript describing this work has been completed and will be submitted shortly.

3. Extended Data Figure 6b-c: although cropped blots are usually not harmful, some of the blots in these two panels were cropped down to a single band, which is less common in publications at this level. It might be good to add a full membrane captures as a supplemental.

We have now included full membrane captures, as suggested (Supplementary Figure 7).

4. Page 7, 1st paragraph: "Moreover, the interaction with ubiquitylated substrate shows clear differences in its binding kinetics, with the K_{on} being significantly faster"  When referring to rate constants, it is common to use small k_{on} , not capitalized K_{on} . It would also be good if actual values will be provided. Additionally, it is not clear how this statement related to the sentence that comes just above it, which describes rather low difference in affinity (~2-fold in term of dissociation constants, K_d ?) between Jarid2 1-530 and the two types of ubiquitylated nucleosomes (wild type vs mutant).

We have included the k_{on} values in the text and modified the sentences to clarify what was meant here.

Reviewers' Comments:

Reviewer #4 (Remarks to the Author):

The authors were reasonably addressed my concerns, either through additional data or through textual adjustments. Both PRC1-PRC2 cross-talk and JARID2-mediated regulation of Polycomb are two hot topics in recent years; so far with only little mechanistic insights. I think this is an important work that brings these two subjects together, while significantly extending our knowledge regarding how PRC2 recognizes the H2AK119ub mark of PRC1 and assigning a novel function for JARID2 in this process. As in many such works, there are some unpolished edges that will likely be subjected to future studies. But the overall quality of the data is sufficiently high to support the central conclusions brought by the authors. I believe that the long revision process of this manuscript, since its initial submission, has gotten to a point where this work is ready to be judged by the community. I am strongly recommending for publication.